# Does Corporate Financialization Affect Corporate Environmental Responsibility? An Empirical Study of China

**Zhenghui Li** [1], **Yan Wang** [2], **Yong Tan** [3] **and Zimei Huang** [2,*]

[1] Guangzhou International Institute of Finance and Guangzhou University, Guangzhou 510006, China; lizh@gzhu.edu.cn
[2] School of Economics and Statistics, Guangzhou University, Guangzhou 510006, China; 2111964052@e.gzhu.edu.cn
[3] Department of Accounting, Finance and Economics, Huddersfield Business School, University of Huddersfield, Queensgate, Huddersfield HD1 3DH, UK; a.y.tan@hud.ac.uk
[*] Correspondence: 2111864007@e.gzhu.edu.cn

**Abstract:** This paper explores the effects and mechanisms of corporate financialization on corporate environmental responsibility (CER), using panel regression and the panel quantile regression model. The data is from 484 Chinese A-share non-financial listed companies, over the period 2008–2015. Some valuable results were achieved, as follows. Firstly, corporate financialization has a significantly negative impact on CER. We attribute this fact to the hard constraint of shareholder value maximization and the soft constraint of CER by taking an extrinsic analysis. Moreover, this negative impact shows heterogeneity. As the CER level increases, the remarkable restraint taken by the corporate financialization on CER is gradually weakened. This results in the corporation aiming not only at the shareholder value maximization, but also at the social effect, rather than only the former. In addition, the effect of the moderating role played by corporate leverage and ownership concentration in the influence of corporate financialization on the CER is captured in different kinds of corporations, while different performances are shown.

**Keywords:** corporate financialization; corporate environmental responsibility; heterogeneity; moderating effect

---

## 1. Introduction

In general, the corporation should create the greatest value for shareholders [1]. However, the corporate environmental responsibility (CER) has increasingly attracted more attention from the public. Corporate financialization is thought of as one of the effective ways to achieve shareholder value maximization [2]. As the corporation faces the risk of being bought out, the administrators have to protect the benefits of shareholders through various short-term financial activities. The corporate investment strategies are compelled towards the objective maximization of shareholder values, which results in the disappearance of the autonomy of productive capital. Even the corporate governance strategies, oriented by shareholder value, realize the maximization of shareholder wealth through changing the corporate incentive mechanism. In order to rapidly increase the corporate value, they have to purse short-term financial profits rather than long-term business strategies. For the benefit of consistency between the corporate administrators and possessors, compensation systems for the administrators depend more heavily on short-term financial performance than product market performance [3]. For instance, provided the administrators are given stock options by which they would be able to push up the share-price, gaining more profits by such means as buying back the

shares, this can lead to popular financial speculation [4,5]. On the other hand, in 2015, the Sustainable Development Goals (SDGs) were adopted as an international sustainable development agenda and will be achieved by 2030, and 6 of the 17 SDG goals directly relate to environmental protection and promotion (goals 6, 7, 12, 13, 14 and 15) [6]. Besides national governments, the achievement of environmental SDGs cannot succeed without a concerted effort by businesses and other stakeholders. With the continuous fast development of the Chinese social economy, damage caused by business activities poses a threat to China's environment [7,8], and will in turn hinder the development of the economy, which also creates obstacles to foreign direct investment [9]. In addition, the enhancement of brand effect of the enterprise is a significant factor, as the corporations prefer to assume more social responsibility and then establish good corporate images. Therefore, the awareness of CER improves significantly. Meanwhile, the literature on financialization from a macro perspective agree with the promotion of financialization for economic development [10–12]. Some classical theories from a micro perspective argue that if the demand for corporate financial assets becomes part of daily transaction demand, it is beneficial to both shareholder value maximization and corporate image enhancement, which results in higher corporate values [13,14].

During recent years, entity corporations gradually reduced their profits, brightly contrasting to the flourishing of financial markets. In order to seek new opportunities for profits, corporations took their first step into financial activities, separating themselves from real economy [15,16]. However, CER is dedicated to improving sustainable development, with the goal of maximizing benefits for stakeholders, which is in conflict with corporate profit maximization through financialization [16,17]. Since the 21st century, entity corporations have gotten into some troubles such as increases in labor cost, excess production capacity, lower external demand, and so on. These existing problems drove them to enter the financial market. With the aim of increasing the profits, they allocated the resources originally used for business operation to financial assets, such that they could earn high returns from the financial markets. The corporate administrators gradually paid their attention to financial departments instead of real departments. Financial assets accounted for more and more of total asset allocation. Especially due to the continuous decline in real economic returns, the corporations invested more funds into the financial departments in order to achieve short-term profits. The result is that the funds available for innovation and production-improvement decreased, slowing down the corporate technical promotion and lowering the operational quality. In a word, corporations deviated themselves from the main businesses [5]. The fulfillment of CER does not have mandatory and unified specification; when corporations focus on the financial sector, corporations tend to neglect CER. However, corporate financialization results in the virtualization of the economy, which increases the outbreak of the financial crisis. In additions, corporate financialization has a negative impact on economic development [4] and employment [18], as well as exacerbating income inequality [2]. Meanwhile, SDGs promote development-oriented policies that support productive activities, decent job creation, entrepreneurship, creativity and innovation. It can be seen that corporate financialization poses a threat to the realization of SDGs, which is not conducive to the sustainable development of economy and society.

From the perspectives of corporate brand and sustainable development, corporations should balance the two objectives of shareholder value and social sustainable development. However, only 10% of the cities of China have achieved an effective balance between environmental protection and economic growth [19]. According to the cost-related theory and value creation theory on the relationship between CER and corporate performance [20], whether CER can benefit companies has raised debates among scholars. Some of the literature argues that undertaking CER has negative impacts on corporate development, due to the fact that CER will increase corporate cost and weaken corporate profitability [21–24]. In addition, CER needs a large amount of funds, which results in the reduction of investment volume to the core business. In general, the corporations are in line with the legal requirement for CER to the extent of the lowest possible cost [25,26]. On the other hand, some of the other literature claims that CER is beneficial for corporate long-term development, as well as

corporate brand awareness. According to the stakeholder theory, CER is able to establish well-deserved reputations among stakeholders. This not only increases the corporate development [27], but also promotes the competitive advantages [28–30]. CER has been recognized globally. In the long term, it will increase corporate values and promote corporate sustainable development [31–34].

The existing works have shown different levels of CER. Different corporations show different balances between the shareholder value and corporate social responsibility. During crises, high corporate social responsibility firms have higher profitability and gross margins, and experience higher sales growth, than other firms [35]. Cai et al. [36] investigated the relationship between CER and risk using a sample of listed companies in the US. They showed that corporations with higher initial CER levels assume lower risks. Li et al. [31] divided CER into three levels: high, middle and low, and found higher corporate values in low and high CER levels. Nollet et al. [37] examined the relationship between corporate social performance and corporate financial performance. Their findings suggest that corporate social responsibility will not have positive impacts on financial performance until it reaches a certain level. Besides, due to heterogeneity in corporations as well as differences in external environment, the enthusiasm of corporations to engage in CER will be influenced. Company characteristics, ownership and contextual factors are the determining factors that influence environmental management and CER practices. For example, unlike private firms, SOEs (State-Owned Enterprises) do not always focus on profit maximization, but also focus on non-profit projects for social benefits as required by government, and are also under greater regulatory pressure from the government [38]. Large companies are normally subject to more public pressure and have more resources available to achieve environmental goals [39]. Firms in environmentally sensitive industries are subject to more scrutiny from government. Liu and Anbumozhi [40], and Brammer and Pavelin [41], found empirically that corporations with large operational scales usually permit higher levels of environmental information disclosure. Zeng et al. [42] reported that state-owned corporations, environmentally sensitive corporations, corporations with more companions in the industry, and corporations with better reputations are more likely to disclose environmental information.

This study focuses on China, an emerging country, which is characterized by joint-stock companies and a concentrated ownership structure. As the influence and restriction of large shareholders increasing, company management gradually loses its independence in making decisions, and shareholder value maximization becomes an important ideology for corporate governance in China. With the decline of profits in the entity sector, it is difficult to safeguard the interests of shareholders and the development of companies. Meanwhile, the continuous improvement of China's capital market and the innovation of financial investment instruments provides more opportunities and temptations for companies to do financial investment, which strengthens the financial investment preference of companies. However, the fast development of the Chinese economy, mainly driven by investments in manufacturing sectors and infrastructures, has caused serious environmental problems [40]. Many corporations appear to be far from environmentally friendly, due to the existing environmental legislative framework and weak enforcement capacity. However, as the largest overall carbon emitter in the world, China needs to demonstrate its global importance in achieving environmental SDGs. Accordingly, China provides an appropriate setting to investigate the relationship between corporate financialization and CER.

The contributions of this paper can be summarized as follows. First, we examine the negative impact of corporate financialization on CER. There is still no consensus regarding the impact of corporate financialization on CER. As the global economic environment evolves, corporations are inclined to maximize the shareholder value, which results in the significantly negative impact of corporate financialization on CER, provided there are no hard constraints on CER. Second, the heterogeneous impact of corporate financialization on CER is evaluated in this paper. Some of the existing literature claims the negative impact of corporate financialization on CER. However, deep analysis of CER levels is lacking. We capture the heterogeneity of the impact of corporate financialization on CER through different quantiles. Finally, we discuss the influencing mechanism, by finding the

heterogeneity of moderating variables playing roles in the impact of corporate financialization on CER in different corporations.

The paper is organized as follows (see Figure 1 for the logical framework). Section 2 presents the examination of the impact of corporate financialization on CER, the purpose of which is to investigate the impact using an econometric model based on the sample data. Section 3 presents the evaluation of the heterogeneity related to the impact of corporate financialization on CER. When taking CER into consideration, we think that corporate culture is diverse, and accordingly the awareness of corporate brand differs, therefore, there would be heterogeneity in the impact of corporate financialization on CER. Section 4 discusses the mechanism of the influence of corporate financialization on CER. Based on the discussion in the sub-samples, we look at whether the impact of corporate financialization on CER would be heterogeneity, through which we will find out the moderate variable in different sub-samples.

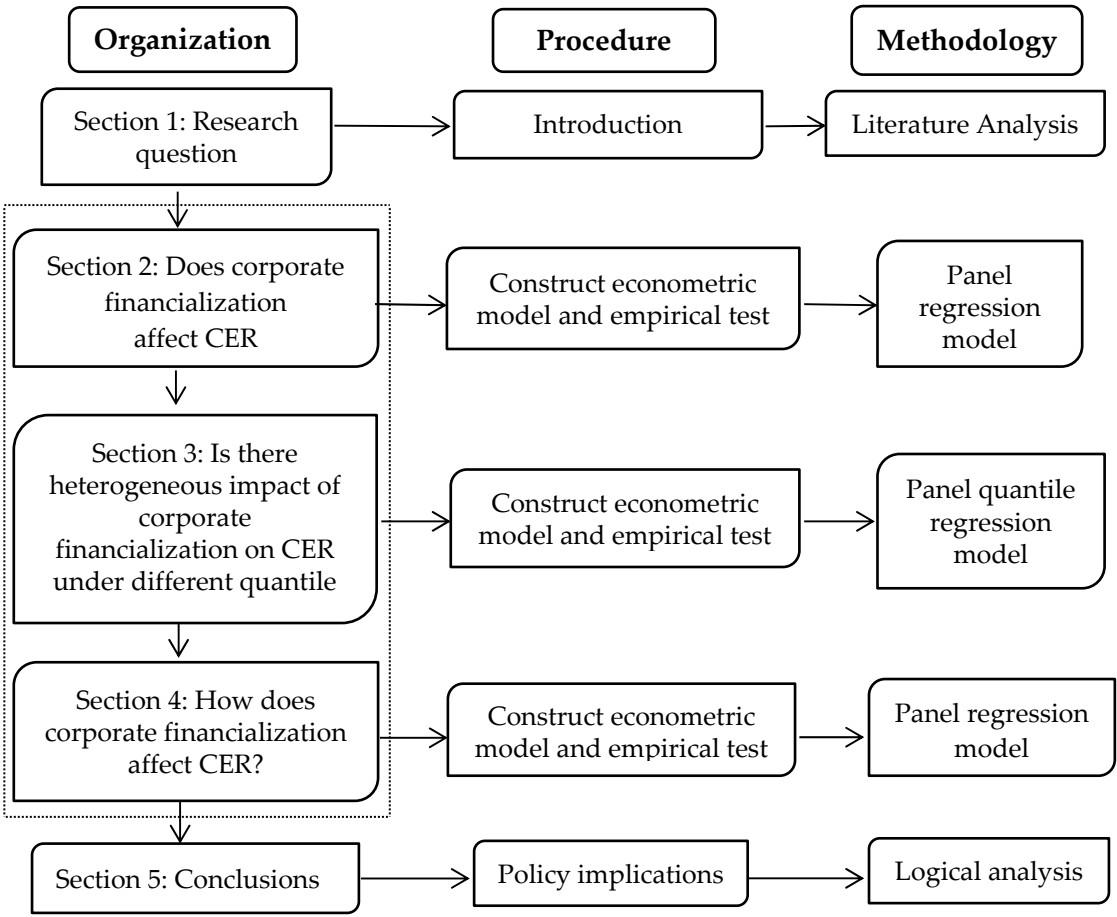

**Figure 1.** The logical framework of this paper.

## 2. The Examination of the Impact of Corporate Financialization on CER

### 2.1. Panel Regression Model

The influence of corporate financialization on CER is mainly attributed to the fact that both of them have different aims to achieve, and CER does have any rigid norm, which leads to the fact that corporations are inclined to maximize the shareholder value. On the one hand, during the corporate operation process, shareholder value maximization has become a dominant ideology for corporate governance [43]. The shareholder ideology has played a key role in the change of corporate financing strategies. The theory of corporate governance advocating shareholder value maximization shows that the integration of benefits for management personnel and the ones for shareholders can improve corporate performance, through giving the management personnel stock options to

incentivize their working effort. Meanwhile, the development of the stock market has a positive impact on foreign portfolio investment [44]. The financialization derived from shareholder value orientation will increase the degree of attention paid by the management personnel to the per-share profits and other performance indicators. Based on this, the purpose of financialization is to orient more effort and consideration towards shareholder benefit, and decrease the corporation's responsibility to other stakeholders. On the other hand, corporate financialization changes the corporate profit mode. Corporations gradually switched their role, from the producers in the product market to the purchasers in the financial markets, through which the resource allocation and business focus have been transferred from operation to the financial area [2]. Although CER can bring good reputation and improve corporate competitiveness in long-term development, which increases corporate value and financial performance, investment in CER has higher levels of cost and will increase the corporate financial burden [23,45]. Therefore, most corporations will select the financial department and give up CER. Meanwhile, financialization squeezes out the investment in research and development at the expense of reducing the level of capital and paying less attention to the long-term planning, whereas the increase in financial investment and arbitrage opportunity takes out the investment in research and development by influencing the incentive mechanism of management personnel [46]. This will reduce the corporate innovation capacity, and makes it detrimental for the corporation to implement environmental responsibility.

The influence of corporate financialization on corporate social responsibility is heterogeneous for different types of corporation, and for corporations with different levels of social responsibility. From the perspective of different types of corporation, due to the fact that different corporations have their own product life cycle and have various financial levels, corporations have diverse aims in the process of asset allocation. For example, for the corporations situated in the early stage of the life cycle, they have a higher liquidity demand, while when the industry turns to the mature stage, corporations have stronger awareness of their brand. From the perspective of the time dimension, the corporate development stage and strategic positioning constrain the aim of capital allocation in the process of asset relocation. For example, shareholder value maximization focuses more on the achievement of short-term aims—they will allocate the capital in the direction which will have the highest level of potential profitability—whereas the establishment of a corporate brand needs more time to accumulate, and both the profit and capital allocation are long-term. In order to investigate the influence of corporate financialization on CER from the perspective of both time and firm, we used the panel data regression model for the analysis. The model is expressed as below:

$$CER_{it} = \beta_0 + \beta_1 FA_{it} + \beta_2 SIZE_{it} + \beta_3 LEV_{it} + \beta_4 \mathrm{Prof}_{it} + \beta_5 OC_{it} + industry + year + \varepsilon_{it}, \qquad (1)$$

where $i$ and $t$ represent firm $i$ and year $t$, CER is the dependent variable reflecting the level of corporate environmental responsibility, and *FA* is the independent variable representing corporate financialization. At the same time, we controlled for a number of variables representing other corporate characteristics that have potential influence on CER, including firm size (SIZE), corporate leverage ratio (LEV), corporate profitability (Prof) and ownership concentration (OC). In addition, in order to alleviate the influence of industry heterogeneity and time on corporate research and innovation activities, we controlled the industry and time fixed effect to eliminate the role of industry characteristics with time invariance, and the time-varying macroeconomic environment.

*2.2. Variables and Data*

2.2.1. Dependent Variable—The Measurement of CER

In order to measure the CER in a systematic way, we adopted the method of Li et al. [31]. We measured CER from five different dimensions, including legal consciousness, social evaluation, eco-friendly production, low-carbon technology and green management [32,47,48], and each dimension had its own indicator, which is summarized in Table 1. For each dimension, we just focused on

whether the firm met a certain condition, but ignored how many times the firm satisfied the conditions. Therefore, we made the assumption that the times that satisfy the condition of each index had no effect on CER. For the indicators, if the answer was yes, firm took the value of 1, and if the answer was no, firm took the value of 0. Further, to keep the direction of all indicators consistent, firms that have been subjected to environmental penalties took the value of 0, and firms that have not been subjected to environmental penalties took the value of 1. In order to ensure the objectivity of the results, all indicators were given the same weight. The sum of the values of indicators under each dimension constitutes the score of each dimension of CER, and the sum of the values of five dimensions constitutes the final CER score.

The value of corporate environmental responsibility at time $t$ for a specific firm $i$ can be expressed as below:

$$CER_{it} = \sum_{k=1}^{5} I_{itk},\qquad(2)$$

where $I_{itk}$ represents the measurement of firm $i$ in year $t$ for dimensions $k$, $CER_{it}$ stands for the final value of corporate environmental responsibility for a specific firm $i$ at a specific year $t$; a higher value of $CER_{it}$ indicates that the corporation had a higher level of corporate environmental responsibility for the specific firm $i$ at the specific year $t$.

**Table 1.** Measurement indicators of CER (Corporate Environmental Responsibility).

| Dimensions | Indicator name |
|---|---|
| $I_1$: Legal consciousness | 1. Whether it follow the guide of sustainable development reporting from Global Reporting Initiative (GRI)<br>2. Whether it discloses environmental and sustainable development<br>3. Whether it is subjected environmental penalties |
| $I_2$: Social evaluation | 1. Whether it receives environmental award<br>2. Whether it has other environmental advantages |
| $I_3$: Eco-friendly production | 1. Whether it adopts circular economy<br>2. Whether it engaged in green production (measurement to decrease three waste) |
| $I_4$: Low-carbon technology | 1. Whether it saves energy<br>2. Whether it generates environmental friendly products |
| $I_5$: Green management | 1. Whether it has been verified by a third party<br>2. Whether it has vision in related to environmental responsibility<br>3. Whether it has environment recognition<br>4. Whether it uses environmental protected offices |

### 2.2.2. Explanatory and Control Variables

With reference to the research of Demir [49], we used the ratio of financial assets to total assets to measure corporate financialization. According to Chinese accounting standards and data availability, we selected monetary capital, trading financial assets, interest receivable, available-for-sale financial assets, and investment in real estate as the indicators to aggregate the total value of the corporate financial assets.

In the process of model-establishment, there are many factors influencing CER. According to relevant theories and the empirical research from scholars around the world [40,41,50,51], there are other factors influencing CER, such as corporate scale and corporate profitability, and others. When examining the effect of corporate financialization on CER, we need to assume that all the other factors are constant, that is, we need to control all the other influencing factors in the quantitative examination, and set the variables as control variables. We selected the control variables referencing the works of other authors. However, this could result in an incomplete selection of control variables. For example, media exposure has an effect on CER in China, but due to the difficulty of data collection, this paper

ignores the impact of media exposure. Based on our systematic summary of the empirical studies combined with the characteristics of Chinese listed companies, we selected four control variables. They are: (1) firm size (SIZE), which was measured by the natural logarithm of total assets; (2) leverage ratio (LEV), which was measured by the ratio of total liabilities to total assets; (3) corporate profitability (Prof), which was measured by ROA (Return On Assets); and (4) ownership concentration (OC), which was measured by the ratio of biggest shareholder holding.

### 2.2.3. Data and Descriptive Statistics

We selected 2008–2015 as the time period for our investigation for the following reasons: first, the Chinese accounting principal was modified in 2006 and 2014. According to its implementation period, relevant financial data is comparable from 2008 to 2016, and there was no fundamental change in the requirements of CER during this period, meaning the study sample does not lose generality. Second, using the Chinese accounting principal, it was found we have lot of missing financial data for 2016.

In terms of the sample type, we employed the A-share non-financial listed companies as our research object, and the data was collected from the China Stock Market & Accounting Research (CSMAR) database and the Chinese Research Data Services Platform (CNRDS). In order to keep the continuity of the sample data, we removed the listed companies of the financial industry and real estate industry, as well as ST (Special Treatment) and PT (Particular Transfer) listed companies, and we also removed the corporations with lots of missing data. We removed the listed companies in the financial industry and real estate industry because the operational business of these businesses belongs to the financial industry and prospective financial industry, and compared to non-financial listed companies, the ratio of their financial assets to their total assets is different, which is not in line with the research objective of this paper. Regarding ST and PT companies, they belong to continuous loss enterprises and do not have sustainable operation capacity, or the general characteristics of financial asset allocation; therefore, we removed them from the sample. Through dealing with these issues, under the time and corporate type constraints, we had in total 484 enterprises in the sample with 3872 observations. The sample covers all 18 non-financial industries (A. Agriculture, forestry, animal husbandry and fishery; B. Mining; C. Manufacturing; D. Electricity, heat, gas and water production and supply; E. Construction industry; F. Wholesale and retail; G. Transportation, storage and postal services; H. Accommodation and catering; I. Information transmission, software and information technology services; L. Leasing and business services; M. Scientific research and technical services; N. Water conservancy, environment and public facilities management; O. Residential services, repairs and other services; P. Education; Q. Health and social work; R. Culture, sports and entertainment; S. Public administration, social security and social organizations; T. international.), except the financial industry and real estate industry. Therefore, the sample in this paper is representative to some extent. After obtaining the sample data, we winsorized all the continuous variables at 1% level to remove the influence of extreme values. The descriptive statistics are presented in Table 2.

**Table 2.** Descriptive statistics.

| Variable | Obs | Mean | Std. Dev. | Min | Max |
|---|---|---|---|---|---|
| CER | 3872 | 4.7422 | 2.5836 | 1 | 12 |
| FA | 3872 | 0.1922 | 0.1299 | 0.0151 | 0.6038 |
| SIZE | 3872 | 9.8424 | 0.6042 | 8.6336 | 11.4806 |
| LEV | 3872 | 0.4930 | 0.1876 | 0.0789 | 0.8859 |
| Prof | 3872 | 0.0497 | 0.0588 | −0.1421 | 0.2358 |
| OC | 3872 | 0.3922 | 0.1628 | 0.0923 | 0.8251 |

Table 2 presents the descriptive statistics of all the variables; overall, the minimum value of CER is 1, the maximum value is 12, while the average value is 4, which indicates that Chinese corporations do not attach great importance to CER. Looking at corporate financialization, the average value of the

ratio of financial assets to total assets is 0.1922, the minimum value is 0.0151 and the maximum value is 0.6038, which suggests that there is quite a lot of difference in the degree of financialization among non-financial corporations, with some of them having a higher degree of financialization. Before examining the impact of financialization on CER, we needed to test the degree of correlation between the variables to make sure they did not suffer from any issue of multicollinearity. Based on this, we calculated the correlation between the variables, the results of which are reported in Table 3.

**Table 3.** Pearson correlation matrix.

| Variables | CER | FA | SIZE | LEV | Prof | OC |
|:---:|:---:|:---:|:---:|:---:|:---:|:---:|
| CER | 1 | | | | | |
| FA | −0.0994 *** | 1 | | | | |
| SIZE | 0.2591 *** | −0.2387 *** | 1 | | | |
| LEV | 0.0866 *** | −0.3443 *** | 0.3894 *** | 1 | | |
| Prof | 0.0569 *** | 0.2793 *** | −0.0545 *** | −0.4682 *** | 1 | |
| OC | 0.0616 *** | −0.1198 *** | 0.2415 *** | 0.0110 | 0.0192 | 1 |

Note: *** $p < 0.01$.

Table 3 shows the Pearson correlation matrix; the results report the correlation between all the variables in the model with CER. We can see that CER is negatively related to corporate financialization, while it is positively related to other control variables. We can also see that all the variables do not suffer from any multicollinearity, and therefore, we can proceed to the regression analysis.

*2.3. Empirical Results*

This part aims to give an exploratory analysis of the impact of corporate financialization on CER. After the stationary test and other related pretreatments of the model, we further investigated the model using the fixed effect and random effect, and fixed effect was found to be superior to the random effect model. Based on this, we used panel data regression with fixed effect for the regression analysis, and at the same time we use the generalized least square estimator (GLS) and maximum likelihood estimator (MLE) for the random effect model, the results of which are reported in Table 4.

Table 4 shows that the coefficient of corporate financialization is −0.868. We also present the results of the random effect model estimated by generalized least square estimator and maximum likelihood estimator; although the assumption and estimation principle for these three methods are different, if there is a stable relationship between variables, the results should be consistent in theory. According to our results, no matter which estimation method is adopted, the coefficient of corporate financialization will be significant and negative at 1% level. The results show that the corporations in China failed to balance the two objectives of shareholder value and social sustainability. The impact of corporate financialization on CER was negative, that is, a higher degree of corporate financialization prevented the implementation of CER. The reason may lie in that during the economic downturn, companies prioritized protecting the interests of shareholders rather than other stakeholders. Due to the decline of profits in the entity sector and the difficulties in safeguarding the interests of shareholders and the development of companies in China, companies allocated assets to the financial sector, seeking short-term high profits. Companies were not willing to bear too much environmental responsibility because of the high expenditures of CER. Therefore, there is a squeezing effect of corporate financialization on CER. With regard to the controlled variables, it is shown that the coefficient of firm size is significant and positive, that is, larger size corporations will be more capable to implement the environmental responsibility. The coefficients of corporate leverage ratio and ownership concentration are significant at 1% level and they are negative, which indicates that corporate leverage ratio and ownership concentration have a constraining effect on the implementation of CER. The impact of profitability on CER is negative but not significant. Table 4 shows that the impact of corporate financialization on CER is significant and negative.

**Table 4.** The influence of corporate financialization on CER.

| Variables | FE | GLS | MLE |
|---|---|---|---|
| FA | −0.858 *** | −1.107 *** | −1.109 *** |
|  | (−0.321) | (−0.404) | (−0.362) |
| SIZE | 1.665 *** | 1.080 *** | 1.064 *** |
|  | (−0.071) | (−0.149) | (−0.123) |
| LEV | −1.131 *** | −0.484 | −0.473 |
|  | (−0.243) | (−0.396) | (−0.309) |
| Prof | −0.212 | −0.392 | −0.392 |
|  | (−0.738) | (−0.862) | (−0.726) |
| OC | −1.038 *** | −0.122 | −0.100 |
|  | (−0.236) | (−0.485) | (−0.373) |
| Constant | −10.500 *** | −7.341 *** | −7.192 *** |
|  | (−0.661) | (−1.386) | (−1.175) |
| industry | Yes | Yes | Yes |
| time | Yes | Yes | Yes |
| Observations | 3872 | 3872 | 3872 |
| R-squared | 0.257 |  |  |
| F-statistic | 131.581 |  |  |
| Logarithmic likelihood |  |  | −7992.51 |

Note: (1) FE represents fixed effect, GLS and MLE are generalized least square estimator and maximum likelihood estimator; (2) Inside of brackets are the standard deviation of the variables; (3) *** $p < 0.01$.

## 3. Heterogeneity Related to the Influence of Corporate Financialization on CER

### 3.1. Panel Quantile Regression Model

There is a heterogeneity related to the degree of environmental information disclosure for different types of corporation, and the influences of corporate financialization on corporations with different levels of environmental responsibility are various. On the one hand, information disclosure of environmental responsibility has two forms: mandatory disclosure and autonomous disclosure. Compared to the corporation with mandatory a disclosure scheme, the corporation with an autonomous disclosure scheme will pay more attention to the interests of stakeholders, and they are more inclined to increase the industrial investment and implement more social responsibility. The corporations with an autonomous information disclosure scheme will communicate information with other stakeholders autonomously based on economic interest, which can channel relevant information related to the corporate core competitiveness and present their own competitive advantages. In comparison, corporations with a mandatory disclosure scheme will communicate the information with other stakeholders based on the legal norms, which leads to incomplete information provision as well as low efficiency. On the other hand, the implementation of CER will bring good reputation for the corporation, which will not only increase the corporation's competitive power, but also improve the corporation's economic development [28]. Furthermore, corporations that more actively implement environmental responsibility will attach more importance to the awareness of environmental protection during their daily operations; low-carbon, environmentally friendly operation modes will reduce the cost of treating large quantities of production pollutants [52]. Therefore, corporations with a higher degree of environmental responsibility will focus more on the long-term sustainable development, and they also pay more attention to the operational business and innovation activities, which reduces the influence of financialization on CER. In comparison, corporations with a lower degree of environmental responsibility will concentration more on short-term profits, and in order to compensate for the reduction in the short-term profits of the real sector, they will increase financial investment to increase the profits, which may be harmful to their competitive power and corporate productivity, and change the sustainable development strategy of the corporations.

In order to test the influence of corporate financialization on CER for the corporations with different levels of the latter, we used the quantile regression model for panel data with fixed effects. Different from the mean regression, the quantile regression model provides a more complete picture of conditional distribution of dependent variables, and the estimation of the quantile regression method is robust to outliers, heteroscedasticity and skewness of dependent variables [53]. Therefore, quantile regression is able to investigate the linear relationship between corporate financialization on CER at different quantiles, through which the marginal effect of the impact of corporate financialization on CER can be analyzed at different quantiles. To a certain extent, this method is able to reflect the information of all sample data, and more realistically reflect the relationship between corporate financialization and CER.

Based on the research objective, we employed the method put by Machado and Silva [54], Method of Moments Quantile Regression (MMQR) with fixed effects. In a conditional location-scale model, the information provided by the conditional mean and the conditional scale function is equivalent to the information provided by regression quantiles, that is, these functions completely characterize how the regressors affect the conditional distribution. Therefore, Machado and Silva estimated quantiles from estimates of the conditional mean and of the conditional scale function. The aggregate data is set up, $\{(CER_{it}, X'_{it})\}'$, among which $CER_{it}$ represents corporate environmental responsibility for a specific firm $i$ at a specific year $t$. $X_{it}$ stands for the corporate financialization, firm size, corporate leverage ratio, corporate profitability and ownership concentration in firm $i$ in year $t$. The random variable CER, whose distribution conditional on a k-vector of covariates X belongs to the location-scale family. Therefore, the estimation of the conditional quantiles $Q_{CER}(\tau|X)$ for a location-scale model of the form

$$CER_{it} = \alpha_i + X'_{it}\theta + \sigma(\delta_i + Z'_{it}\gamma)U_{it}, \tag{3}$$

with $P\{\sigma(\delta_i + Z'_{it}\gamma) > 0\} = 1$. The parameters $(\alpha_i, \delta_i)$, $i = 1, \dots, n$, capture the firm $i$ fixed effect, and Z is a $k$-vector of known differentiable transformations of the components of X. The sequence $\{X_{it}\}$ is i.i.d. for any fixed firm $i$ and independent across time $t$. $U_{it}$ is i.i.d. across firm $i$ and time $t$, statistically independent of $X_{it}$, and normalized to satisfy the moment conditions:

$$E(U) = 0 \, , E(|U|) = 1 \tag{4}$$

Model (3) implies that

$$Q_{CER}\left(\tau|X'_{it}\right) = \alpha_i + X'_{it}\theta + \sigma(\delta_i + Z'_{it}\gamma)q(\tau), \tag{5}$$

in the case where $\sigma(\cdot)$ is the identity function and $Z = X$, the quantiles simplify to

$$Q_{CER}\left(\tau|X'_{it}\right) = (\alpha_i + \delta_i q(\tau)) + X'_{it}(\theta + \gamma q(\tau)) \tag{6}$$

Therefore, we specify the panel quantiles function for quantile $\tau$ as follows:

$$Q_{CER_{it}}\left(\tau|\alpha_i, \varepsilon_{it}, X'_{it}\right) = \alpha_i + \varepsilon_{it} + \theta_{1\tau}FA_{it} + \theta_{2\tau}SIZE_{it} + \theta_{3\tau}LEV_{it} + \theta_{4\tau}Prof_{it} + \theta_{5\tau}OC_{it}, \tag{7}$$

where $i$ and $t$ represent firm $i$ and year $t$, *FA* stands for corporate financialization, *SIZE* is firm size, *LEV* represents corporate leverage ratio, *Prof* is corporate profitability, and *OC* indicates ownership concentration. The scalar coefficient $\alpha_i(\tau) = \alpha_i + \delta q(\tau)$ is the quantile-$\tau$ fixed effect for firm $i$, or the distributional effect at $\tau$. The distributional effect represents the effect of time-invariant individual characteristics, which are allowed to have different impacts on different regions of the conditional distribution of CER. The fact that $\int_0^1 q(\tau)d\tau = 0$ implies that $\alpha_i$ can be interpreted as the average effect for firm $i$.

### 3.2. Empirical Results

In order to test the influence of corporate financialization on CER and find out whether it would be different for corporations with different degrees of environmental responsibility, panel data quantile regression was used to analyze five quantile levels at 10%, 25%, 50%, 75% and 90%. The results are reported in Table 5.

**Table 5.** Results on the panel data regression analysis.

| Variables | 0.1 | 0.25 | 0.5 | 0.75 | 0.9 |
|---|---|---|---|---|---|
| FA | −1.044 * | −0.957 ** | −0.850*** | −0.760 * | −0.681 |
| | (−0.626) | (−0.434) | (−0.313) | (−0.398) | (−0.56) |
| SIZE | 1.919 *** | 1.800*** | 1.654 *** | 1.531 *** | 1.423 *** |
| | (−0.141) | (−0.098) | (−0.0708) | (−0.0898) | (−0.126) |
| LEV | −1.497 *** | −1.326 *** | −1.116 *** | −0.940 *** | −0.785 * |
| | (−0.5) | (−0.347) | (−0.25) | (−0.318) | (−0.447) |
| Prof | 0.532 | 0.184 | −0.242 | −0.602 | −0.917 |
| | (−1.458) | (−1.01) | (−0.729) | (−0.926) | (−1.304) |
| OC | −1.365 *** | −1.212 *** | −1.025 *** | −0.867 *** | −0.729 * |
| | (−0.463) | (−0.321) | (−0.231) | (−0.294) | (−0.414) |
| Constant | 0.328 ** | 0.329 *** | 0.331 *** | 0.332 *** | 0.333 ** |
| | (−0.153) | (−0.106) | (−0.0766) | (−0.0973) | (−0.137) |
| Observations | 3872 | 3872 | 3872 | 3872 | 3872 |

Notes: Standard errors are in parentheses; *** $p < 0.01$, ** $p < 0.05$, * $p < 0.1$.

The regression analysis from Tables 4 and 5 uses different regression methods, and the sign of the coefficients of the variables are consistent in general, but as the quantile of CER changes, the degree of influence and significance of the variables are heterogeneous. For all the quantile levels, the coefficient of corporate financialization is negative. If we look at the quantile levels on a one-by-one basis, the coefficients increase as the quantile increases; at 75% quantile level, the coefficient is higher by 0.284 compared to the one at the 10% quantile level, while the coefficient at the 90% level is the largest, at −0.681, however, it is not significant. At the 50% quantile level, the coefficient of corporate financialization is −0.850. Compared to the common panel data regression analysis, the coefficient of which is −0.858, these two estimations reached similar results, that is, using the panel data regression analysis and panel data quantile analysis at 50% quantile level, the coefficients are very similar, which shows the results are robust. The above results mean that as the degree of CER undertaken by the corporation increases, the constraint effect of corporate financialization decreases. With regard to the control variables, the firm size, corporate leverage ratio and ownership concentration are in line with the ones from Table 4, all of which are significant, while profitability is not significant. As the quantile level increases, the positive impact of firm size on CER becomes smaller, and the constraint effect of corporate leverage ratio and ownership concentration on CER becomes weaker. To be more specific, different levels of CER will have different impacts on corporations, and there would be significant differences in balancing the shareholder value and CER among corporations. For corporations with a lower degree of the implementation of CER, stakeholders pay less attention to CER, which results in firms not being able to get enough profit from CER activities. Therefore, corporations orient to the financial department, which earns corporations more profit through different financial channels, and decreases the investment in environment aspects. However, corporations with a higher level of CER would be brought good reputation and competitive advantages, which increases their corporate value. Therefore, corporations would pay attention to the implementation of CER, and the squeezing effect of corporate financialization on CER would become weaker.

## 4. The Mechanism of the Influence of Corporate Financialization on CER

### 4.1. The Heterogeneity in Different Types of Corporation

Although the above results can be used to understand as a whole the influence of corporate financialization on CER, it ignores the influence on different types of corporation. Industry characteristics, ownership characteristics and firm size are the important factors that influence the CER [40,50,55]. The crowding-out effect of the influence of corporate financialization on CER is various across firms within different types. First, different firms have different degrees of emphasis and enthusiasm relating to the implementation of CER. Due to the fact that the government and public have different degrees of monitoring and supervision, compared to the less environmentally sensitive industries, environmentally sensitive industries face more strict regulation as they cause serious damage to the environment. Therefore, firms in more environmentally sensitive industries will pay more attention to the implementation of the CER. Compared to the private ownership enterprises, on account of the specific characteristics of shareholders and particularity of political representation, together with the pressure from relevant legal regulations, the state-owned listed enterprises should attach more importance to environmental information disclosure and implementing the social responsibility [42]. The operational behavior of large enterprises has a great influence on society, and they also face high pressure from political regulation and supervision from the public. In order to establish a good corporate impact and provide a good example, large enterprises will actively implement relevant environmental responsibility. Meanwhile, large enterprises have relatively adequate capital and human resources, which gives them a higher ability to implement their environmental responsibility [39,40]. Secondly, because of the financing constraints, state-owned enterprises and large enterprises have an advantage in getting bank loans—they face weaker financing constraints—whereas the private ownership enterprises and small enterprises with higher financing constrains are more sensitive to the cost of implementing environmental responsibility. CER is more easily influenced by the corporations themselves and financial institutions. Therefore, it would be necessary to divide the whole sample into different sub-samples for further investigation and discussion.

Based on the theoretical analysis above, we divided the whole sample into sub-samples according to the industry characteristics, ownership characteristics and firm size. According to the industry characteristics, we divided the sample into less environmentally sensitive industries and more environmentally sensitive industries (According to the A guide to the classification of listed companies modified by China Securities regulatory commission in 2012, the List of listed companies in the classified management of environmental verification industry, and the Environmental information disclosure guidelines for listed companies formulated in 2008 by the Ministry of environmental protection, the current paper includes the following industries as heavy polluting industries: mining and washing of coal industry; extraction of petroleum and natural gas; processing of ferrous metals ores; non-ferrous metals mining and dressing; textile industry (leather, fur, feather and its products) and shoemaking; papermaking and paper products; petroleum processing and coking and nuclear fuel processing; manufacturing of chemical materials and products; pharmaceutical industry; chemical fiber manufacturing industry; manufacture of non-metallic mineral products; ferrous metal smelting and rolling processing industry; nonferrous metal smelting and rolling processing industry; metal product industry; production and supply of electric power and heat power (totally 16 classifications)), while according to the share ownership, we divided the sample into state-owned enterprises and private ownership enterprises. Finally, according to the size of operations, we divided the sample into small corporations, with the firm size smaller than the average value, and large corporations, with one larger than the average value. We used the panel data regression analysis for the estimation, the results of which are reported in Table 6.

**Table 6.** The heterogeneity of the influence of corporate financialization on CER.

| Variables | Less Environmentally Sensitive | More Environmentally Sensitive | State-Owned | Private Ownership | Small | Large |
|---|---|---|---|---|---|---|
| FA | −1.209 *** | −0.458 | −0.365 | −1.791 *** | −1.442 *** | 0.291 |
|  | (−0.388) | (−0.565) | (0.406) | (0.531) | (0.395) | (0.548) |
| SIZE | 1.802 *** | 1.395 *** | 1.609 *** | 1.913 *** | 1.412 *** | 1.569 *** |
|  | (−0.095) | (−0.106) | (0.081) | (0.167) | (0.166) | (0.128) |
| LEV | −1.157 *** | −1.212 *** | −1.193 *** | −0.924 ** | −0.952 *** | −1.676 *** |
|  | (−0.309) | (−0.392) | (0.287) | (0.470) | (0.309) | (0.396) |
| Prof | 2.083 ** | −3.172 *** | −0.937 | 1.282 | 0.401 | −1.769 |
|  | (−0.975) | (−1.141) | (0.938) | (1.286) | (0.941) | (1.188) |
| OC | −1.599 *** | −0.025 | −0.907 *** | −1.690 *** | −1.133 *** | −0.923 ** |
|  | (−0.301) | (−0.387) | (0.292) | (0.439) | (0.317) | (0.359) |
| Constant | −11.780 *** | −7.922 *** | −10.030 *** | −12.640 *** | −7.934 *** | −9.548 *** |
|  | (−0.886) | (−0.985) | (0.747) | (1.537) | (1.552) | (1.282) |
| industry | Yes | Yes | Yes | Yes | Yes | Yes |
| time | Yes | Yes | Yes | Yes | Yes | Yes |
| Observations | 2360 | 1512 | 2674 | 1159 | 2133 | 1739 |
| R−squared | 0.267 | 0.233 | 0.264 | 0.240 | 0.190 | 0.263 |
| F−statistic | 86.852 | 49.931 | 95.262 | 35.613 | 20.914 | 31.631 |

Notes: Standard errors are in parentheses; *** $p < 0.01$, ** $p < 0.05$.

Table 6 reports the influence of financialization on CER in different types of corporations. For the analysis of less environmentally sensitive industry and more environmentally sensitive industry, the coefficients of corporate financialization are −1.209 and −0.458; the former is significant at 1% level, while the latter is insignificant. The results show that corporate financialization significantly constrains the implementation of CER in more environmentally sensitive industries, but the effect is insignificant in less environmentally sensitive industries. For the regression analysis of state-owned enterprises and private ownership enterprises, the coefficients of corporate financialization are −0.365 and −1.791, and the private ownership enterprises are significant at 1%, while state-owned enterprises are insignificant. For the regression analysis of large and small enterprises, the coefficients of corporate financialization are −1.442 and 0.291, and the small enterprises are significant at 1% level, while the large enterprises are insignificant. This is because China's CER is still in its early stages and the supervision system and related laws are immature. Corporations in more environmentally sensitive industries face more strict supervision, because of the damage their production activities cause to the environment. State-owned corporations and large corporations face high pressure from political regulation and supervision from the public, which makes them pay more attentions to CER. However, as CER is not mandatory, less environmentally sensitive industry, private ownership corporations and small corporations pay less attention to CER. In addition, private ownership corporations and small corporations often face high financing constraints, leading to more financial speculation. Therefore, as enterprises attach varying importance to CER and the different degree of need for capital, the influence of corporate financialization on CER is different in different kinds of corporation, and the constraint effect of corporate financialization on CER is more reflected in less environmentally sensitive industry, private ownership corporations and small corporations.

## 4.2. The Moderating Effect between Corporate Financialization and CER in Different Types of Corporation

Leverage ratio is the main indicator for measuring the ability of the economic entity for debt repayment, and it is also the indicator reflecting the risk level of debt. A higher leverage ratio indicates corporations have a higher level of risk. The agency theory argues that higher levels of corporate leverage ratio will increase the agency cost, derived from the conflict of interest between managers and owners. In order to prevent the possibility related to the transfer of wealth from the creditors to the shareholders, creditors ask for increases in the level of information disclosure, meaning corporate leverage ratio has a positive influence on environmental information disclosure [56,57]. In addition, because of higher risk, corporations with higher leverage ratios will increase the investment for undertaking social

responsibilities in order to divert investors' attention from the corporate risk. Therefore, corporations with higher levels of leverage ratio will be more oriented towards implementing more CER and avoiding corporate financialization. Corporate ownership concentration reflects the control of large shareholders over the company; a higher ownership concentration will strengthen the shareholders' control and more effectively monitor the manger's decision-making [58]. Large shareholders play different roles in different situations (supervision effect and plunder effect). With higher ownership concentration, large shareholders have a stronger supervisory role over managers [59], which makes large shareholders actively participate in the enterprise's investment decisions and alleviate the internal inefficient behavior. Therefore, large shareholders will be more considerate of corporate sustainable development; they will increase the implementation of CER, and avoid the corporation's shortsightedness. Meanwhile, with higher ownership concentration, the predatory motivation of shareholders will be strengthened, and the large shareholders will empty the wealth of enterprises at the expense of the interests of the minority shareholders [60]. Large shareholders hope to receive profits in short-term, and they will reduce the investment related to the environment and increase the investment in the financial department. Thus, this paper investigates the moderating effect of corporate leverage ratio and ownership concentration between financialization and CER.

Based on the above analysis, we added the interaction term between financialization and corporate leverage ratio, as well as the interaction between financialization and ownership concentration, to our baseline panel data regression model (model 1) to examine the moderating effect. In order to increase the explanatory capacity, we centralized the data of financialization, corporate leverage ratio and ownership concentration, and then estimate the parameters using the panel data regression analysis, the results of which are reported in Table 7.

**Table 7.** The moderating role of the influence of corporate financialization on CER.

| Variables | Less Environmentally Sensitive | More Environmentally Sensitive | State-Owned | Private Ownership | Small | Large |
|---|---|---|---|---|---|---|
| c_FA | −1.313 *** | 0.535 | −0.580 | −1.646 *** | −1.440 *** | 0.0623 |
| | (0.400) | (0.659) | (0.439) | (0.563) | (0.412) | (0.562) |
| c_LEV | −1.075 *** | −1.247 *** | −1.088 *** | −0.926 ** | −0.952 *** | −1.566 *** |
| | (0.310) | (0.393) | (0.289) | (0.472) | (0.311) | (0.390) |
| c_OC | −1.578 *** | −0.035 | −0.859 *** | −1.671 *** | −1.134 *** | −0.909 ** |
| | (0.300) | (0.384) | (0.292) | (0.429) | (0.318) | (0.357) |
| FA*LEV | −2.877 * | 5.952 *** | −3.019 * | 2.391 | 0.035 | −6.599 ** |
| | (1.701) | (2.270) | (1.727) | (2.435) | (1.703) | (2.584) |
| FA*OC | −3.023 | 5.414 ** | −2.808 | 0.474 | 0.016 | −6.233 ** |
| | (2.270) | (2.418) | (2.118) | (3.183) | (2.321) | (2.975) |
| SIZE | 1.801 *** | 1.437 *** | 1.592 *** | 1.887 *** | 1.412 *** | 1.509 *** |
| | (0.096) | (0.107) | (0.082) | (0.172) | (0.168) | (0.133) |
| Prof | 2.142 ** | −3.228 *** | −0.812 | 1.321 | 0.401 | −1.772 |
| | (0.984) | (1.137) | (0.947) | (1.293) | (0.950) | (1.190) |
| Constant | −13.230 *** | −8.957 *** | −10.961 *** | −13.750 *** | −9.069 *** | −10.240 *** |
| | (0.925) | (1.060) | (0.808) | (1.638) | (1.576) | (1.381) |
| industry | Yes | Yes | Yes | Yes | Yes | Yes |
| time | Yes | Yes | Yes | Yes | Yes | Yes |
| Observations | 2360 | 1512 | 2674 | 1159 | 2133 | 1739 |
| R−squared | 0.269 | 0.238 | 0.265 | 0.241 | 0.190 | 0.268 |
| F-statistic | 68.191 | 37.260 | 70.511 | 26.421 | 14.973 | 24.652 |

Notes: Standard errors are in parentheses; *** $p < 0.01$, ** $p < 0.05$, * $p < 0.1$.

Table 7 shows that, regarding the regression analysis of less environmentally sensitive industry and more environmentally sensitive industry, the coefficients of interactions between corporate leverage ratio and financialization are −2.877 and 5.952, and they are significant at 10% and 1% levels, respectively. The interaction between ownership concentration and financialization is only significant in the more environmentally sensitive industry, the coefficient of which is 5.414. In terms of the regression analysis of state-owned and private ownership enterprises, the interaction between

corporate leverage ratio and financialization is only significant in the state-owned enterprises, the coefficient of which is −3.019. Regarding the regression analysis of large and small enterprises, the interaction between corporate leverage ratio and financialization, as well as the interaction between ownership concentration and financialization, are significant in large corporations, the coefficients of which are −6.599 and −6.233, while they are insignificant in small corporates. The findings suggest that the leverage ratio plays a positive moderating role in the relationship between financialization and CER in more environmentally sensitive industry, while a negative effect was shown for less environmentally sensitive industry, as well as state-owned enterprises and large enterprises. We explain this finding by the fact that more environmentally sensitive industry faces stricter regulation, and when the corporates leverage ratio increases, they will implement more CER in order to divert investors' attention from the corporate risk. However, there is no strict requirement for corporate implementation of CER for less environmentally sensitive industry or large corporations, and high leverage is a serious problem in state-owned enterprises in China. When the corporate leverage ratio increases, they may ignore their environmental responsibilities, taking into consideration the fact that the inputs in implementing CER will increase corporations' operational cost. In some of the literature, it is also surprising to find a negative relationship between corporate leverage and environmental information disclosure [41,61]. Therefore, corporate leverage will intensify the constraining effect of financialization on CER. In addition, the ownership concentration will have a positive moderating effect on the relationship between financialization and CER in the more environmentally sensitive industries, but the effect is negative in large enterprises. This is attributed to the fact that the supervision effect of major shareholders is greater than the plunder effect in the more environmentally sensitive industries. Large shareholders pay more attention to the implementation of CER for corporate long-term development, which will alleviate the negative impact of financialization on CER. On the contrary, the plunder effect of large shareholders may play a dominant role in decision-making. Based on the consideration of their self-interest, large shareholders hope to receive profit in the short-term, tending to increase the investment in their financial department and reduce the degree of CER, which exasperates the negative influence of financialization on CER. Therefore, for different types of enterprises, the leverage ratio and the ownership concentration have moderating effects on the influence of financialization on CER, and the effect is heterogeneous.

## 5. Conclusions

This study uses a sample of A-share non-financial listed companies, over the period 2008–2015, to investigate the impact of corporate financialization on CER. Using panel data regression analysis and panel data quantile regression analysis, we derived the following findings:

First, the impact of corporate financialization on CER is significant and negative. There is still no consensus regarding the impact of corporate financialization of CER, but this paper, using Chinese listed companies as the example, further confirms the significant and negative impact of corporate financialization on CER. This negative impact is attributed to: first, corporations orient their focus toward shareholder value maximization; second, there are no hard constraints to CER.

Second, there is a heterogeneity regarding the impact of corporate financialization on CER. The heterogeneity is mainly reflected in different quantiles. Higher levels of financialization will impede the implementation of CER, however, as the implementation of environmental responsibility increases, the negative influence of corporate financialization on CER becomes weaker gradually. This is attributed to the improvement in corporate culture, for example, the enhancement of corporate brand awareness. As the level of CER improves, the corporate value is changed from the pursuing of shareholder value maximization to the dual purpose of shareholder value maximization and social effect maximization.

Third, corporate leverage and ownership concentration play a moderating effect in the influence of corporate financialization on CER, and the impact of financialization on CER is different among various types of corporation. On the one hand, for companies in less environmentally sensitive

industries, private ownership and small corporations, the impact of financialization on CER is significant and negative, whereas, for corporations in more environmentally sensitive industries, state-owned and large corporations, the impact of financialization on CER is insignificant. On the other hand, corporate leverage and ownership concentration plays a moderating role in the influence of corporate financialization on CER, and the moderating effect varies in different kinds corporation. In the more environmentally sensitive industries, the moderating effect of corporate leverage is positive, whereas, in the less environmentally sensitive industries, state-owned and large corporations, the moderating effect is negative. The ownership concentration plays a positive moderating role in more environmentally sensitive industries, while this role is negative for large corporations.

In summary, the empirical findings of the study offer important practical implications for managers, government, policy makers, suppliers and creditors. First, because of the negative influence of corporate financialization on CER, enterprise managers need to pay attention to how to balance the two objectives of shareholder value maximization and sustainable development. Meanwhile, corporations could reduce the leverage ratio and adjust the share concentration in order to alleviate the negative influence of corporate financialization on CER. Second, the government, who is the primary stakeholder in Chinese companies, should realize the current circumstances of monopoly and extremely high profit in the financial industry, and effectively direct the financial capital to support corporate development and promote the implementation of the SDGs, as well as complete the environment monitoring mechanism. Third, for different types of corporates, the policy makers could formulate structured and differentiated policies in accordance with the heterogeneity of corporation, in particular, special attention should be paid to the corporations in the less environmentally sensitive industries, private ownership corporations and small corporations. In addition, suppliers and creditors should raise awareness of sustainability and require companies to disclose more information about environmental protection. Only with the efforts of all stakeholders can sustainable development be realized. Just like the COVID-19 epidemic, it not only needs the mutual assistance of governments, the efforts of medical workers and the financial support of large enterprises, but it also needs the cooperation and support of every person.

**Author Contributions:** Conceptualization, Z.L., Y.W., Y.T. and Z.H.; methodology, Y.W. and Z.H.; software, Y.W.; validation, Z.L., Y.W., Y.T. and Z.H.; formal analysis, Z.L., Y.W., Y.T. and Z.H.; investigation, Z.L. and Y.T.; resources, Z.L. and Y.T.; data curation, Y.W. and Z.H.; writing—original draft preparation, Z.L., Y.W., Y.T. and Z.H.; writing—review and editing, Z.L., Y.W., Y.T. and Z.H.; visualization, Y.W. and Z.H.; supervision, Z.L., Y.W., Y.T. and Z.H.; project administration, Z.L.; funding acquisition, Z.L. All authors have read and agreed to the published version of the manuscript.

**Funding:** This research was funded by the NATIONAL OFFICE of PHILOSOPHY and SOCIAL SCIENCES, grant number 19BGL050.

**Acknowledgments:** Authors would like to thank Guangzhou University for sponsoring this research.

**Conflicts of Interest:** The authors declare no conflict of interest.

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
