# Peer review of "Does Corporate Financialization Affect Corporate Environmental Responsibility? An Empirical Study of China"

_sustainability, doi:10.3390/su12093696_

Round 1

Reviewer 1 Report

First of all, this manuscript is very well organized and written accordingly to the submission guidelines of this journal, sustainability. This paper explored one of the most significant but somewhat new concept Corporate Environmental Responsibility (CER). The authors’ scholarly attempts concerning its relationships on value maximation and corporate financialization are in line with scientific research methodology. That is, the procedural details of data collection, analysis, representation and other important aspects such as validation and reliability evidence gathering are properly described. Moreover, one of the unique and advantageous aspect of this study is in-depth discussion on CER and its potential impact on financialization of corporations. Therefore, potential readers of this journal would acquire new knowledge from consuming this study.  

Author Response

Reply to the reviewer 1 comments for the manuscript “Does Corporate Financialization Affect Corporate Environmental Responsibility? An Empirical Study of China” Dear Editors and Reviewers: Thank you very much for your letter and comments concerning our manuscript entitled“Does Corporate Financialization Affect Corporate Environmental Responsibility? An Empirical Study of China”. The comments are valuable and very helpful for us to revise and improve our paper, as well as the important guiding significance to our researches. We have studied comments carefully and have made correction which we hope meet with approval. Revised portion are marked in red in the paper. The main corrections in the paper and the responds to the reviewer 1 comments are as following: Comments and Reply: First of all, this manuscript is very well organized and written accordingly to the submission guidelines of this journal, sustainability. This paper explored one of the most significant but somewhat new concept Corporate Environmental Responsibility (CER). The authors’ scholarly attempts concerning its relationships on value maximation and corporate financialization are in line with scientific research methodology. That is, the procedural details of data collection, analysis, representation and other important aspects such as validation and reliability evidence gathering are properly described. Moreover, one of the unique and advantageous aspect of this study is in-depth discussion on CER and its potential impact on financialization of corporations. Therefore, potential readers of this journal would acquire new knowledge from consuming this study. Reply: Thanks for the reviewer’s comments. We appreciate for Reviewers’ encouragement earnestly.

Reviewer 2 Report

The selected topic sounds promissing, but the performed research, analysis with excessively complicated formulas and rather unexpected conclusions are below expectations. Therefore, the paper needs signficiant improvements and modifications, inlcuding English proofreading, in order to be considered for the publication.

My strongest objections:

  1. the background, including referencing, needs to be expanded;
  2. the research sample needs to be better explained (we know the time period and the type of companies, but we do not know about the industries, etc.) - is it homogenous? respresentative?
  3. how was the data collected? We know Indicators from Table 1 (188 line), but how were these indicators applied? Who and how was made the "grading"? Was this objective?
  4. the methodology needs to be better explained - these formulas look overcomplicated and detached from reality;
  5. the results must be supported by a well argued discussion. I do not see the links between data and propositions;
  6. my biggest objection: How the authors came to the conclusion that CSR should normatively be regulated by the government? From where could they take the certainty that the government should decide for businesses about their sustainability approach and CSR strategy? This sounds to me as a total denial of the multi-stakeholder model and of private ownership.
  7. the English must be improved - proofreading is a must!!!

Author Response

Reply to the reviewer 2 comments for the manuscript “Does Corporate Financialization Affect Corporate Environmental Responsibility? An Empirical Study of China”

Dear Editors and Reviewers:

Thank you very much for your letter and comments concerning our manuscript entitled“Does Corporate Financialization Affect Corporate Environmental Responsibility? An Empirical Study of China”. The comments are valuable and very helpful for us to revise and improve our paper, as well as the important guiding significance to our researches. We have studied comments carefully and have made correction which we hope meet with approval. Revised portion are marked in red in the paper. The main corrections in the paper and the responds to the reviewer 2 comments are as following:

Comments and Reply:

  1. The background, including referencing, needs to be expanded;

Reply: Thanks for the reviewer’s comments. In order to complete the structure of the article, we added the background about the characteristics of Chinese ownership, the reason for the financialization of enterprises and the problem of fulfilling CER in Chinese enterprises (line 101-114). And we have also added two references which support our idea of the measurement of CER (line 194, reference 35, 36). The content of the modification has been marked in red in the article.

Reference:

  1. Kolk, A. The social responsibility of international business: From ethics and the environment to CSR and sustainable development. J. World Bus. 2016, 51 (1), 23-34. Doi: 10.1016/j.jwb.2015.08.010.
  2. Kim, H.;K. Park;D. Ryu. Corporate Environmental Responsibility: A Legal Origins Perspective. J. Bus. Ethics 2017, 140 (3), 381-402. Doi: 10.1007/s10551-015-2641-1.

  1. The research sample needs to be better explained (we know the time period and the type of companies, but we do not know about the industries, etc.) - is it homogenous? respresentative?

Reply: The sample used in this paper covers all 18 kinds of non-financial industries, including 62 kinds of industries. The 18 kinds of non-financial industries aren’t homogenous and guarantee the universality of the sample to some extent. The details of the industries have been added in the paper and marked red (line 251-253).

  1. How was the data collected? We know Indicators from Table 1 (188 line), but how were these indicators applied? Who and how was made the "grading"? Was this objective?

Reply: The information needed in the CER indicator is obtained from the CSR report issued by the listed company, which had been verified by a third party and the data can be collected from China Stock Market &Accounting Research database and Chinese Research Data Services Platform. The indicators of CER were converted into the quantitative ones based on whether the firm meets a certain condition. We reinterpreted the measurement of CER in detail and remarked red in the paper. (line 198-204)

  1. The methodology needs to be better explained - these formulas look overcomplicated and detached from reality;

Reply: According to the suggestion from the reviewer, we have better explained the methodology. In order to make the methodology employed in this paper easier to understand, we have made some modifications to the paper (line 343-345, line 351). We change the formula (3) (5) (6) and add the explanation about the transformation condition between formula (5) and (6), which have been marked red in the paper.

  1. The results must be supported by a well argued discussion. I do not see the links between data and propositions;

Reply: According to the suggestion from the reviewer, a discussion had been added in the paper. First, we gave the reason why we focus on China. (line 101-114) Then, we did some modification on discussion from every empirical result. (line 291-294, line 382-389, line 437-441, line 489-496 and line 501-508) The content of the modification has been marked in red in the article.

  1. My biggest objection: How the authors came to the conclusion that CSR should normatively be regulated by the government? From where could they take the certainty that the government should decide for businesses about their sustainability approach and CSR strategy? This sounds to me as a total denial of the multi-stakeholder model and of private ownership.

Reply: According to the heterogeneity test results of the impact of financialization on CER in different kinds of corporation, it shows that financialization has a significant impact on CER in the more environmentally sensitive industry, state-owned and large corporations. In addition, government regulation and the requirements of relevant laws are important factors affecting CER, which has been confirmed in the previous studies. Therefore, based on the fact that China's listed companies are still at a low level in fulfilling CER, this paper argues that government regulation will bring pressure to enterprises, which is conducive to the fulfillment of CER, and can alleviate the inhibitory effect of financialization on CER. But we are not repudiating the multi-stakeholder model and private ownership. In order to better illustrate the conclusion, some modifications have been made to the paper and marked red. (line 542-550)

  1. The English must be improved - proofreading is a must!

Reply: According to the suggestion from the reviewer, the English has been proofreading by native English speaker.

Reviewer 3 Report

Enclosed

Author Response

Reply to the reviewer 3 comments for the manuscript “Does Corporate Financialization Affect Corporate Environmental Responsibility? An Empirical Study of China”

Dear Editors and Reviewers:

Thank you very much for your letter and comments concerning our manuscript entitled“Does Corporate Financialization Affect Corporate Environmental Responsibility? An Empirical Study of China”. The comments are valuable and very helpful for us to revise and improve our paper, as well as the important guiding significance to our researches. We have studied comments carefully and have made correction which we hope meet with approval. Revised portion are marked in red in the paper. The main corrections in the paper and the responds to the reviewer 3 comments are as following:

Comments and Reply:

  1. Problem and aim formulated properly. No hypotheses are formulated. In general formal terms it is correct. However, the paper would have a greater scientific value in a hypothesis (hypotheses) were formulated – in reference to the earlier results, referring to specificity of Chinese market. However, I do not treat it as necessity. The Authors should be at least aware of that. Formally the concept of papers is correct. The Authors are familiar with the theoretical knowledge and results of the previous research. It is an advantage of the paper, 71-98.

Reply: Thanks for your concerns and comments. No hypotheses are formulated in our paper, but we express our views on the relationship between corporate financialization and CER through theoretical analysis. (line 154-163, line 321-328, line 395-396, line 452-453, line 459-461 and line 463-465)

  1. Since the paper is prepared according to the dominating scheme its main weakness is almost textbook-like case. It is a typical situation when specialists in econometrics (macro & micro scale) built the mathematical models of the phenomena. In the macro scale the results of such modelling – the dependent variables are usually (not always) explainable. However, in the micro scale (e.g. company as here), the main challenge and weakness can be expressed with the following question: What is the meaning of the CER indicator (ratio, etc.?). What is the dimension (in physical and/or in financial terms)? Is it dimensionless? It looks that in the equation 1 the control variables are mixed (fractions, i.e. proportions + an absolute value, i.e. the size of the company)

Reply: The value of the indicator of CER is an absolute value, which take the value of 0 or 1. The dimensions of CER are in physical terms and dimensionless. It may be a weakness that the control variables are mixed, but this case also can be seen in other works. The size of the company, which is always measured by the natural logarithm of total assets, is an important factor affecting CER.

  1. Looking at the description of the equation 1, the Authors want to explain two things – the dimensions of the indicator CER and the control variables determining this indicator. They face two challenges:
  2. Defining the “dimensions” of the CER indicator – and quantifying them.
  3. Identification of the control variables.

Reply:

  1. In order to measure the CER in a systematic way, we measured CER from five different dimensions including legal consciousness, social evaluation, eco-friendly production, low-carbon technology and green management. And we quantified indicator of each dimensions by using 0 and 1.
  2. According to relevant theories and the empirical research, firm size, leverage ratio, corporate profitability and ownership concentration have an important effect on CER. Therefore, we selected them as the control variables in this paper and explained how to measure them.

  1. This is always the main challenge of that kind of works. Their authors, usually proficient in statistics must make several simplifying assumptions. It allows to show the “scientific value” of the research by applying more or less advanced statistical, or in a broader sense mathematical methods. My questions in such case:
  2. How the qualitative characteristics were converted into the quantitative ones.
  3. Are the Authors aware of limitations of the quantification, or in a broader sense, operationalization.

Reply:

  1. In fact, CER were converted into the quantitative ones based on whether the firm meets a certain condition. For example, firms that have been subjected to environmental penalties take the value of 0, and firms that have not been subjected to environmental penalties take the value of 1. We reinterpreted the measurement of CER in detail and marked in red in the paper. (line 198-204)
  2. The limitation of the quantification in this paper is that all indicators of CER are given the same weight, which may prevent us from accurately measuring the value of CER. However, it avoids subjectivity of empowerment and reflects the value of CER of the enterprise to some extent.

  1. The Authors’ ideas of “dimensions” are collected in Table 1 (dimensions). This is the crucial issue which has been omitted by the Authors. As to have the full picture, the following questions should be answered:
  2. Are the “dimensions” qualitative or quantitative?
  3. If quantitative – what type of Stevens’ scaling can be applied in their definitions?
  4. Are the “dimensions” commensurable?

Reply:

  1. The dimensions of CER are qualitative.
  2. b. As the dimensions of CER are qualitative, Stevens’ scaling that including nominal scale, ordinal scale, interval scale and ratio scale can’t be applied in their definitions.
  3. The “dimensions” are commensurable. For the following reason: the level of CER is measured by five dimensions. Each dimension has its own indicators, which are quantified as 0 or 1 based on the same principle that whether the firm meets a certain condition. Therefore, the score of each dimension can be added. The sum of the values of indicators under each dimension constitutes the score of each dimension of CER, and the final CER score is the sum of the values of five dimensions.

  1. Obviously, complete answers to these questions never can be given but in the era of increasing awareness of a constructivist (interpretative) character of numbers in social research (social constructs), the Authors of this type of works should at least show that they are aware of that problem. If they make any simplifying assumptions in answering to the above questions (5 a.b.c), explanations for this assumptions should be presented. In addition to the explanation of the models, this problem should be signaled in the limitations.

Reply: The level of CER needs to be measured by five dimensions. For each dimensions, we just focus on whether the firm meets a certain condition but ignore how many times the firm satisfy the conditions. Therefore, we make an assumption that the times that satisfy the condition of each index has no effect on CER. The explanation of limitations of such a selection had been added and marked red. (line 195-198)

  1. Similar doubts can be expressed in reference to the control variables. In this case, however, the Authors very correctly show the roots of their control variables in the works of other authors. However, at least a short explanation of limitations of such a selection should be also added.

Reply: We selected the control variables referencing the works of other authors. However, this may results in an incomplete selection of control variables. For example, media exposure has an effect on CER in China, but due to the difficulty of data collection, this paper ignores the impact of media exposure. The explanation of limitations of such a selection had been added and marked red. (line 223-226)

  1. My another comment concerns the assumption that maximization of the shareholder value is the main aim of the company. Firstly, it can be seen that this principle can be seen as one of the reasons of the problems in the US economy (not only). The Authors know about the limitations and they mention also the role of the stakeholders. It is well-documented with the relevant sources. However, a short comment about the limitations could be helpful. It is not needed. It is just a comment addressed to the Authors.

Reply: Thanks for your concerns and comments. As we focus on the relationship between corporate financialization and CER, we didn’t mention the limitations of the goal of the maximization of the shareholder value on corporate governance. However, we analyzed why the goal of the maximization of the shareholder value affects the relationship between corporate financialization and CER. (in section 2.1)

  1. Selection of the sample can be also disputable but the Authors’ arguments are correct. However, a question can be asked. Maximization of shareholders value is commonly accepted as the aim of company but it has very strong link with the US corporate ownership, capital market, financial markets, etc. Another question: To what extent the ideas deriving from the USA are relevant to the A-share non-financial listed companies in China? Going further: The nuclear fuel processing industry is quoted. Although I do not know too much about this sector of industry in China but I know something about specificity of that industry (I have double background in physics and in management). I am slightly puzzled how the ideas of financialization born in the USA can be adequate to that sector in China.

As to be understood properly, I do not take into account any political factors, only business management. Therefore the Authors should also add an explanation why the equivalence between the ideas deriving from management of the US companies are applicable in China?

Reply: According to the suggestion from the reviewer, in order to better understand, we added the background about the characteristics of Chinese ownership, the reason for the financialization of enterprises in China. The content of the modification has been marked in red in the article. (line 104-114)

  1. Data collection, processing and interpretations of results are a strong formal/methodological feature of the paper. I find the entire modelling as a sign of good quality of the team. The approach to heterogeneity is innovative.

Reply: Thanks for your comments.

  1. Due to obvious reasons, I am not able to prove all numerical results but the statistical processing is correct.

Reply: Thanks for your comments. We have studied comments carefully and have made correction which we hope meet with approval. Revised portion are marked in red in the paper.

Round 2

Reviewer 2 Report

The authors have partially followed some of my recommendations, but I must underline that these improvements are just minor and do not resolve my most significant objections. Adding two references can not take care of the foundation and do not overcome a big academic deficit. Even more importantly, the methodology is still confusing. The biggest issue is the lack of proper analysis and discussion leading to the conclusions. I am glad that the authors do not repudiate the multi-stakholder model, but actually it should be the other way around. See SDGs!!! We all must work eagerly towards the multi-stakehodler CSR/sustainability approach. Especially, considering the current situation (COVID-19)!

Author Response

Dear Editors and Reviewers: Thank you for your letter and for the reviewers’ comments concerning our manuscript entitled “Does Corporate Financialization Affect Corporate Environmental Responsibility? An Empirical Study of China”. Those comments are all valuable and very helpful for revising and improving our paper, as well as the important guiding significance to our researches. We have studied comments carefully and have made correction which we hope meet with approval. Revised portion are marked in red in the paper. The main corrections in the paper and the responds to the reviewers’ comments are as following: Reviewer 2: Comments and Suggestions for Authors The authors have partially followed some of my recommendations, but I must underline that these improvements are just minor and do not resolve my most significant objections. Adding two references can not take care of the foundation and do not overcome a big academic deficit. Even more importantly, the methodology is still confusing. The biggest issue is the lack of proper analysis and discussion leading to the conclusions. I am glad that the authors do not repudiate the multi-stakeholder model, but actually it should be the other way around. See SDGs!!! We all must work eagerly towards the multi-stakeholder CSR/sustainability approach. Especially, considering the current situation (COVID-19)! 1. Adding two references can not take care of the foundation and do not overcome a big academic deficit. Reply: Thanks for the reviewer’s comments. According to the SDGs and relevant literature, we added the urgency of environmental sustainable development and the negative role of corporate financialization in enterprise sustainable development, so as to demonstrate the contribution and significance of this research (line 46-52, line 75-83). Meanwhile, we analyzed the impact of CER on enterprises, the importance of CER in different types of enterprises and the actual situation of Chinese enterprises fulfilling their environmental responsibilities in a more specific way (line 84-89, line 111-117, line 132-135). We also added ten references to support our ideas. All modifications have been remarked red in paper. 2. Even more importantly, the methodology is still confusing. Reply: We are sorry that we didn’t give a perfect answer. Thus, we try to explain the methodology. In this paper, we explore the effects of corporate financialization on CER from the perspective of micro-econometrics. The research paradigm of micro-econometrics is to construct the corresponding econometrics model through theoretical analysis and obtain important conclusions from the empirical results. At the same time, we provide references that are similar to our research methodology. In this paper, we use panel regression model and quantile regression model for panel data with fixed effects, which refers to the Machado and Silva Method of Moments Quantile Regression (MMQR) with fixed effects. In order to better explain the MMQR method, we added the main idea of MMQR in the paper and remarked red. (line 368-373) References: 1. Ortas, E.;I. Gallego-Alvarez;I. Álvarez Etxeberria. Financial Factors Influencing the Quality of Corporate Social Responsibility and Environmental Management Disclosure: A Quantile Regression Approach. Corp. Soc. Responsib. Environ. Manag. 2015, 22 (6), 362-380. Doi: 10.1002/csr.1351. 2. Ike, G. N.;O. Usman;S. A. Sarkodie. Testing the role of oil production in the environmental Kuznets curve of oil producing countries: New insights from Method of Moments Quantile Regression. Sci. Total Environ. 2020, 711, 10. Doi: 10.1016/j.scitotenv.2019.135205. 3. Arora, P.;R. Dharwadkar. Corporate Governance and Corporate Social Responsibility (CSR): The Moderating Roles of Attainment Discrepancy and Organization Slack. Corp. Gov. 2011, 19 (2), 136-152. Doi: 10.1111/j.1467-8683.2010.00843.x. 3. The biggest issue is the lack of proper analysis and discussion leading to the conclusions. Reply: Thanks for the reviewer’s comments. On the basis of theoretical analysis, we analyzed the discussion in a more specific way based on our empirical results. We explained why there is a squeezing effect of corporate financialization on CER in China and the heterogeneity in different level of CER and in different kinds of corporations and remarked red. (line 313-322, line 415-424, line 472-484, line 532-539, line 544-551) 4. I am glad that the authors do not repudiate the multi-stakeholder model, but actually it should be the other way around. See SDGs!!! We all must work eagerly towards the multi-stakeholder CSR/sustainability approach. Especially, considering the current situation (COVID-19)! Reply: Thanks for the reviewer’s comments. In order to improve the conclusion, based our empirical results, we analyzed what should each stakeholder do, including managers, government, policy makers, suppliers and creditors. The modifications have been remarked red in paper. (584-602) We tried our best to improve the manuscript and made some changes in the manuscript. We appreciate for Reviewer’s warm work earnestly, and hope that the correction will meet with approval. Once again, thank you very much for your comments and suggestions.
